# Knowledge, attitude and practices towards COVID-19 preventive measures among adults in Bhutan: A cross-sectional study

**Tshering Yangzom●\*◉, Tshering Cheki●◉, Nirmala Koirala‡, Dipsika Rai‡**

Department of Nursing Services, Jigme Dorji Wangchuck National Referral Hospital, Thimphu, Bhutan

◉ These authors contributed equally to this work.

‡ NK and DR also contributed equally to this work.

* jampelyang17@gmail.com

**Data Availability Statement:** All relevant data are within the paper and its Supporting Information files.

## Abstract

The COVID-19 pandemic posed a major global health challenge. Preventive measures against the spread of COVID-19 require the involvement of all sections of society. Knowledge and attitude towards COVID-19 preventive measures influence human practices. We describe the knowledge, attitude and practice (KAP) of COVID-19 preventive measure in Bhutan. This was a cross-sectional survey using multistage-cluster sampling involving participants from all 20 districts of the country. The knowledge was assessed using 12 items, attitude using 6 Likert items and practice using 10 items. There was total of 1708 respondents. The mean knowledge was 10.7, (SD = 1.5; range 0–12); 86.38% had good knowledge, 10.95% had average knowledge, 2.69% had poor knowledge. The common sources of knowledge were television (84.9%) and family and friends (74.7%). Those younger than 30 years were associated with good knowledge. The fear of contracting COVID-19 was reported by 96% and 86.4% agreed that appropriate preventive measures can help control the spread of COVID-19. Nearly all the respondents (97%) wore mask while going out and majority practiced good hand hygiene (87.9%) and proper cough etiquette (84.1%). The knowledge on COVID-19 preventive measures was good and the majority held positive attitudes and practices.

## Introduction

A new coronavirus (COVID-19) not earlier identified in humans emerged in Wuhan, China in December 2019 [1–3]. Coronaviruses are a group of viruses belonging to the family of Coronaviridae, which infect both animals and humans. According to the World Health Organization (WHO), viral diseases continue to emerge and represent a serious issue to public health. COVID-19 transmission continue across the globe and 630,601,291 confirmed cases and 6,583,588 deaths as of 8th November, 2022 [4].

The disease primarily cause respiratory illness ranging from asymptomatic individuals, mild infections to severe forms including acute respiratory distress symptoms and death [5].

**Funding:** The study was supported by UNICEF Bhutan from the fund released for conduct of Advocacy/Awareness on COVID-19 vaccination to the Vaccine Preventable Disease Program (VPDP), Department of Public Health, Ministry of Health. The fund was solely to facilitate enumerator recruitment hence, the funders had no role in study design, data collection and analysis, decision to publish, or preparation of the manuscript.

**Competing interests:** The author(s) declare(s) that there is no conflict of interest.

Individuals with pre-existing illnesses (such as diabetes, hypertension, malignancy, kidney disease, cardiac and lung diseases), unvaccinated children and elderly are vulnerable for severe disease and mortality [6, 7].

It is mostly transmitted through droplet, contact and fomites [5, 8]. Taking simple precautions, such as using a face mask, practicing good hand hygiene (hand wash or hand rub), avoiding crowds and gatherings, keeping physical distance, observing good coughing etiquette, cleaning and disinfecting surfaces, and reporting flu-like symptoms are key measures to slow and prevent the spread of the disease [1, 5, 8].

In Bhutan, the surveillance for COVID-19 was activated by mid-January 2020 [9] people entering the country via air and land were screened for fever and flu-like symptoms. The first COVID-19 case in Bhutan was detected in March 2020 in a tourist visiting Bhutan [10]. Later with increase COVID-19 cases, nation-wide lockdown and mandatory quarantine for those traveling from high-risk areas were implemented. As of 11 November 2022, Bhutan had reported 62,430 COVID-19 cases and 21 deaths [11].

While COVID-19 is a new disease, knowledge, attitude and risk perception are key factors associated with the adoption of preventive measures and control of infectious disease [1, 12]. It is known that levels of knowledge, attitude and practice (KAP) play a vital role in practicing COVID-19 preventive measures [1, 3]. Bhutan is a small country with a population of 0.7 million situated in the eastern Himalayas. It has an overall literacy rate of 71.4% and adult literacy rate of 66.6% [13]. All levels of healthcare including testing and treatment of COVID-19 were provided free of cost by the government. This study was conducted to describe the KAP on COVID-19 preventive measures among Bhutanese population.

## Methods

### Study design, study setting and study population

This cross-sectional survey was conducted in 20 districts of Bhutan. The target population was Bhutanese people who were ≥18 years at the time of data collection and agree to participate in the research were surveyed. In 2017, the adult population (>20 years) in the country was 469,442 [14].

### Sample size and sampling

For this study, we assumed that 50% of the respondents would have good knowledge and 50% would have positive attitude and practice towards COVID-19 preventive measures. The sample size was calculated for proportions considering 95% confidence interval, margin of error 0.05 with design effect of 2 to address the issue of cluster sampling. Assuming an expected 90% response rate, the calculated sample size was 1720. A multi-stage cluster sampling method was used for recruitment of participants.

### Study instrument

While there are a variety of instruments used for the assessment of KAP on COVID-19, none were suitable for Bhutan. We therefore, designed a questionnaire for the purpose for this study (S1 File). It was initially prepared in English and then translated into national language Dzongkha.

The tool consisted of four sections: seven items on socio-demographic characteristics of participants and one item on the sources of information, twelve items for testing knowledge, six items to assess participants' attitude towards COVID-19 preventive measures and ten items to assess the participants' COVID-19 preventive practices.

A panel of six experts assessed the face and content validity of the instrument. The scale-level content validity index was 0.98 for knowledge component, 0.93 for attitude component and 0.97 for practice component. The item-level content validity index ranged from 0.8 to 1.0 and content validity ratio ranged from 0.7 to 1.0. The instrument was pretested among 30 participants through convenience sampling. The internal consistency reliability (Cronbach's α) for the instrument was 0.88.

### Data collection

There were 30 trained enumerators who collected the data on Epicollect5. The enumerators were fluent in English, Dzongkha and local dialect of the region. The participants were clearly informed about the background and objectives of the study. The survey was conducted between 9th January to 28th February 2022.

### Statistical analysis

The data were exported to and analyzed in STATA 13.1. Continuous variables are summarized using mean and standard deviation (SD) and categorical variables are summarized using frequency and percentage.

For the assessment of knowledge, the correct response to an item was assigned 1 point, while an incorrect or "don't know" response was assigned 0 points. The range of knowledge score ranged from 0–12 with higher indicating better knowledge about the COVID-19 preventive measures. Those with knowledge score 10–12 points were categorized as having good knowledge, 7–9 points as average knowledge and 0–6 points as poor knowledge. Factors associated with good knowledge vs average and poor knowledge was tested using logistic regression. Findings with $p < 0.05$ were considered significant.

For the assessment of attitude, a five-point Likert scale was used. The practice points are described as frequencies and percentages.

### Ethical clearance

The study was approved by the Research Ethics Board of Health, Ministry of Health, Thimphu (REBH/Approval/2021/142 dated 20/12/2022). The administrative clearance was obtained from the Policy and Planning Division, Ministry of Health, survey clearance was obtained from National Statistics Bureau, Royal Government of Bhutan and site clearance was obtained from district administrators. An online informed consent was obtained by the enumerators before proceeding with the survey. Consent was made available in English and the national language Dzongkha.

## Results

### Demographic characteristics

There were 1708 participants in the survey (response rate 99.3%). The majority were female (52.9%) and married (72.8%), more than half were younger than 40 years (54.5%) and the majority lived in rural area (74.2%). The details of the socio-demographic characteristics of the respondents are shown in Table 1.

### Knowledge regarding COVID-19 preventive measures

The mean knowledge regarding COVID-19 preventive measure was 10.7 (SD±1.51, range: 0–12). The majority had good knowledge (86.4%), 10.9% had average knowledge and 2.7% had poor knowledge. Respondents had good knowledge on preventive measures such as hand

**Table 1. Socio-demographic characteristics of the respondents of the knowledge, attitude and practice on COVID-19 preventive measures survey in Bhutan, January–February, 2022 (n = 1708).**

| Characteristics | n | (%) |
|---|---|---|
| Age (years) | | |
| 18–29 | 449 | (26.3) |
| 30–39 | 481 | (28.2) |
| 40–49 | 334 | (19.6) |
| 50–59 | 238 | (13.9) |
| >60 | 206 | (12.1) |
| Gender | | |
| Male | 805 | (47.1) |
| Female | 903 | (52.9) |
| Marital status | | |
| Unmarried | 339 | (19.8) |
| Married | 1243 | (72.8) |
| Divorced | 57 | (3.3) |
| Widow(er) | 69 | (4.0) |
| Level of education | | |
| Cannot read and write | 597 | (35.0) |
| Non-formal education | 109 | (6.4) |
| Monastic education | 65 | (3.8) |
| Primary | 170 | (10.0) |
| Secondary | 524 | (30.7) |
| Diploma | 73 | (4.3) |
| University education or more | 170 | (10.0) |
| Occupation | | |
| Not-employed | 517 | (30.3) |
| Student | 147 | (8.6) |
| Private sector | 189 | (11.1) |
| Government service | 232 | (13.6) |
| Others[1] | 623 | (36.5) |
| Settlement type | | |
| Rural | 1268 | (74.2) |
| Urban | 440 | (25.8) |
| Level of monthly income (Nu) [2] | | |
| <10,000 | 1088 | (63.7) |
| 10,001–20,000 | 268 | (15.7) |
| 20,001–30,000 | 223 | (13.1) |
| 30,001–40,000 | 83 | (4.9) |
| >40,001 | 46 | (2.7) |

[1]Other included farmer, business, armed personnel, monk, driver and carpenter

[2]US dollar 1 = Ngultrum (Nu) 77 in January 2022

hygiene (98.5%), use of face mask (98.0%), physical distancing (97.1%) and avoidance of crowd (97.1%). Respondents had poor knowledge on surface contamination (56.8%). Most respondents knew that people who are either infected or had contact with infected person should be immediately isolated and quarantined. A great majority of the respondents (90.7%) answered that vaccination is important to prevent COVID-19. The details of responses to knowledge assessment are shown in Table 2.

**Table 2. Response to knowledge questions on COVID-19 preventive measures among participants surveyed for the knowledge, attitude and practices survey in Bhutan, January–February, 2022 (n = 1708).**

| Item no. | Knowledge Items | True | | False | | Don't know | |
|---|---|---|---|---|---|---|---|
| | | n | (%) | n | (%) | n | (%) |
| K1 | Wearing a facemask can effectively prevent transmission of virus. | 1673 | (98.0) | 16 | (0.9) | 19 | (1.1) |
| K2 | Hand hygiene (washing or sanitizing) can prevent getting COVID-19 infection. | 1682 | (98.5) | 4 | (0.2) | 22 | (1.3) |
| K3 | Sneezing or coughing into your arm/elbow can help prevent the spread of the virus. | 1521 | (89.1) | 71 | (4.2) | 116 | (6.8) |
| K4 | You should maintain a safe distance of at least one meter between yourself and others. | 1659 | (97.1) | 14 | (0.8) | 35 | (2.0) |
| K5 | Virus can be transferred by shaking hands and touching your face (eyes, nose, and mouth.) | 1572 | (92.0) | 14 | (0.8) | 122 | (7.1) |
| K6 | You should avoid going to crowded places (e.g., restaurants, religious gatherings, bars, etc.) | 1658 | (97.1) | 10 | (0.6) | 40 | (2.3) |
| K7 | You should minimize or avoid taking public transportation. | 1562 | (91.5) | 25 | (1.5) | 121 | (7.1) |
| K8 | The virus can stay on objects for a few days to weeks. | 970 | (56.8) | 75 | (4.4) | 663 | (38.8) |
| K9 | Stay home and self-isolate (avoid going to work, school and social gatherings) even if you have minor symptoms such as cough, headache and mild fever. | 1584 | (92.7) | 36 | (2.1) | 88 | (5.2) |
| K10 | Those confirmed/suspected infections and primary contacts of the COVID-19 patients should be immediately isolated and quarantined. | 1654 | (96.8) | 8 | (0.5) | 46 | (2.7) |
| K11 | Children and young adults need not take COVID-19 preventive measures. | 1236 | (72.4) | 328 | (19.2) | 144 | (8.4) |
| K12 | Vaccination is important to prevent COVID-19 infection. | 1550 | (90.7) | 43 | (2.5) | 115 | (6.7) |

The common sources of information about COVID-19 preventive measures were television (84.9%), family and friends (74.7%), social media (69.6%) and healthcare professionals (60.8%). The details of the sources of information are shown in Fig 1.

Compared to respondents in the age group 18–29 years, those in the age-groups of 30–39 years (adjusted OR 0.61, 95% CI 0.38–0.99, p = 0.044), 40–49 years (adjusted OR 0.55, 95% CI 0.32–0.96, p = 0.034) and 50–59 years (adjusted OR 0.45, 95% CI 0.45–0.83, p = 0.010) were

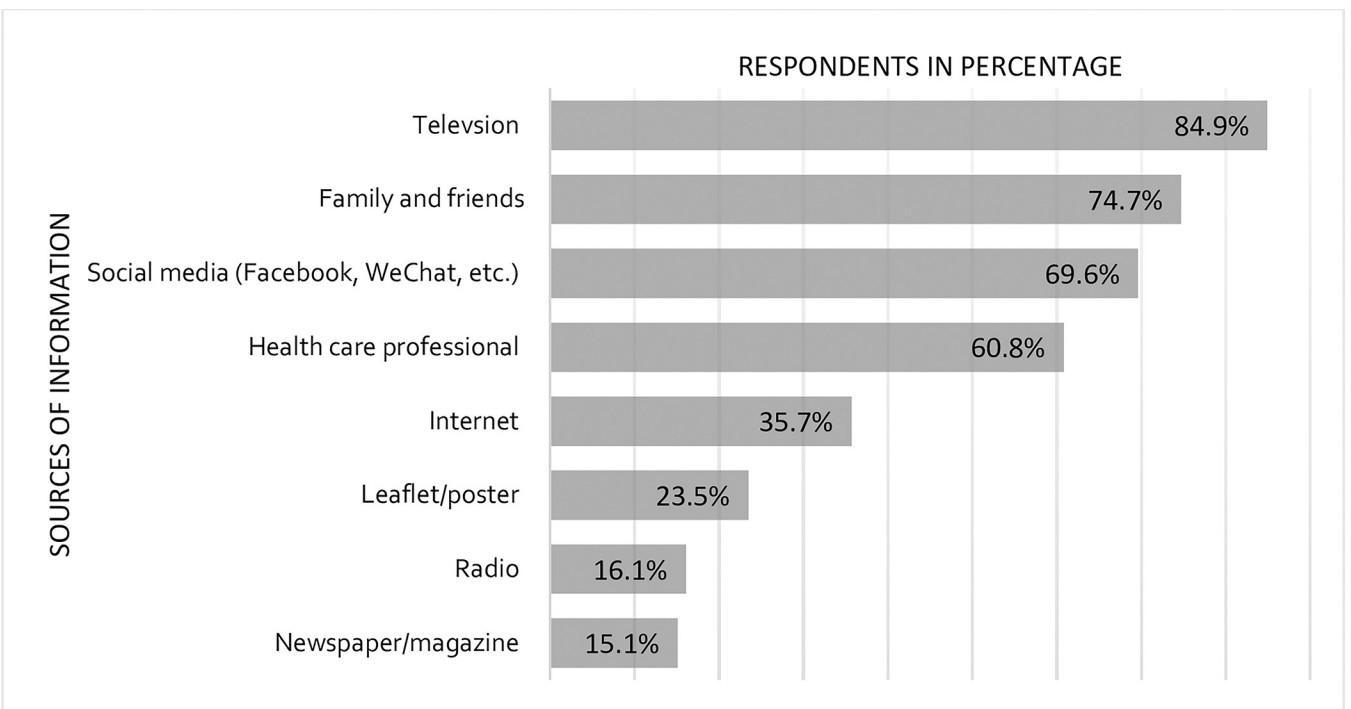

**Fig 1. Sources of information among respondents of the knowledge, attitude and practices on COVID-19 preventive measures survey in Bhutan, January–February, 2022.**

less likely to have good knowledge. Compared to those who could not read and write, those with secondary level of education were less likely to have good knowledge (OR 0.57, 95% CI 0.36–0.92, p = 0.022). Compared to those in government service, those who were unemployed (adjusted OR 2.55, 95% CI 1.21–5.39, p = 0.014), those working in the private sector (adjusted OR 3.59, 95% CI 1.70–7.59, p = 0.001) and others (adjusted OR 2.31, 95% CI 1.11–4.81, p = 0.025) were likely to have good knowledge. The details of factors associated with good knowledge on COVID-19 preventive measures is shown in Table 3.

**Table 3. Factors associated with good knowledge regarding COVID-19 preventive measures among participants surveyed for the knowledge, attitude and practices survey in Bhutan, January–February, 2022 (n = 1708).**

| Variable | Adjusted odds ratio (95% CI) | p value |
|---|---|---|
| Age (years) | | |
| 18–29 | | |
| 30–39 | 0.61 (0.38–0.99) | **0.044** |
| 40–49 | 0.55 (0.32–0.96) | **0.034** |
| 50–59 | 0.45 (0.25–0.83) | **0.010** |
| ≥60 | 0.76 (0.41–1.39) | 0.366 |
| Gender | | |
| Male | | |
| Female | 1.10 (0.82–1.49) | 0.521 |
| Marital status | | |
| Unmarried | | |
| Married | 1.31 (0.79–2.19) | 0.298 |
| Divorced | 1.05 (0.39–2.86) | 0.919 |
| Widow(er) | 2.19 (0.99–4.81) | 0.052 |
| Level of education | | |
| Cannot read and write | | |
| Non formal education | 0.55 (0.29–1.04) | 0.067 |
| Monastic education | 0.45 (0.17–1.18) | 0.103 |
| Primary | 1.06 (0.65–1.70) | 0.822 |
| Secondary | 0.57 (0.36–0.92) | **0.022** |
| Diploma | 0.38 (0.12–1.19) | 0.096 |
| >University degree | 0.61 (0.30–1.26) | 0.181 |
| Occupation | | |
| Government service | | |
| Not-employed | 2.55 (1.21–5.39) | **0.014** |
| Student | 1.44 (0.54–3.81) | 0.465 |
| Private sector | 3.59 (1.70–7.59) | **0.001** |
| Others[1] | 2.31 (1.11–4.81) | **0.025** |
| Settlement | | |
| Rural | | |
| Urban | 0.94 (0.65–1.35) | 0.729 |
| Level of monthly income (Nu)[2] | | |
| <10,000 | | |
| 10,001–20,000 | 0.76 (0.47–1.23) | 0.262 |
| 20,001–30,000 | 0.80 (0.45–1.42) | 0.446 |
| 30,001–40,000 | 1.19 (0.56–2.53) | 0.650 |
| >40,001 | 0.50 (0.15–1.71) | 0.271 |

[1]Other included farmer, business, armed personnel, monk, driver and carpenter

[2]US dollar 1 = Ngultrum (Nu) 77 in January 2022

**Table 4. Attitudes towards COVID-19 preventive measures among participants surveyed for the knowledge, attitude and practices survey in Bhutan, January–February, 2022 (n = 1708).**

| Item No. | Attitude points | Agree | | Not Sure | | Disagree | |
|---|---|---|---|---|---|---|---|
| | | n | (%) | n | (%) | n | (%) |
| A1 | I pay close attention to the spread of COVID-19 in the country. | 1554 | (91.0) | 124 | (7.3) | 30 | (1.8) |
| A2 | COVID-19 is an important health problem in our country. | 1123 | (94.8) | 66 | (3.9) | 22 | (1.3) |
| A3 | My life has been disturbed by the COVID-19. | 1274 | (74.6) | 204 | (11.9) | 230 | (13.5) |
| A4 | Following the COVID-19 preventive protocols is important in controlling the pandemic. | 1477 | (86.5) | 214 | (12.5) | 17 | (1.0) |
| A5 | I fear of contracting COVID-19. | 1640 | (96.0) | 23 | (1.3) | 45 | (2.6) |
| A6 | Mandatory quarantine for travelers coming from high-risk areas is an effective preventive measure. | 1644 | (96.3) | 56 | (3.3) | 8 | (0.5) |

## Attitude of general public towards COVID-19 preventive measures

A great majority (94.8%) reported that COVID-19 is an important health problem for Bhutan and showed interest (91.0%) in knowing about the situation of spread of COVID-19. Almost two thirds (74.6%) reported that their lives have been disturbed by the pandemic. The majority (86.4%) agreed that COVID-19-appropriate measures can prevent the spread of the infection and 96.4% were in support of the government measure of travel-related quarantine. The fear of COVID-19 infection was reported in 96%. The details of the assessment of attitude towards COVID-19 preventive measures is shown in Table 4.

## Practice of COVID-19 preventive measures

The mean COVID-19 practice score of the respondents was 32.4 (SD±5.28, range: 10–40), suggesting overall 81.1% correct rate of practice. Nearly all (97.1%) wore face mask while going out. The vast majority (87.4%) of the respondents were adhering to good hand hygiene either by washing hand or using hand sanitizer. Greater portion (84.1%) of the respondents reported of practicing proper cough etiquette. Little more than two-thirds of the respondents reported of avoiding public transport (78.6%), maintenance of physical distance (76.2%), avoidance of shaking hands (76.2%), and not going to crowded places (75.2%). In addition, 73.6% avoided going out unnecessarily while symptomatic and 72.7% visited flu clinic on worsening of the symptoms. The least practiced preventive measure was disinfection of the frequently touched surfaces (43.9%). The details of the assessment of practice of COVID-19 preventive measures are shown in Table 5.

**Table 5. COVID-19 preventive practices among participants surveyed for the knowledge, attitude and practices survey in Bhutan, January–February, 2022 (n = 1708).**

| Item No. | Practice points | Always | | Often | | Sometimes | | Never | |
|---|---|---|---|---|---|---|---|---|---|
| | | n | (%) | n | (%) | n | (%) | n | (%) |
| P1 | I wear a facemask when I go out. | 1504 | (88.1) | 154 | (9.0) | 50 | (2.9) | 0 | (0.0) |
| P2 | I wash my hands or use hand sanitizer. | 1081 | (63.3) | 421 | (24.6) | 204 | (11.9) | 2 | (0.1) |
| P3 | I cover my mouth and nose with my bent elbow or a tissue when I cough or sneeze. | 1002 | (58.7) | 434 | (25.4) | 243 | (14.2) | 24 | (1.4) |
| P4 | I maintain a distance of at least one meter when meeting others. | 757 | (44.3) | 544 | (31.9) | 378 | (22.1) | 29 | (1.7) |
| P5 | I avoid shaking hands or touching my face (eyes, nose, and mouth). | 874 | (51.2) | 426 | (25.0) | 302 | (17.7) | 101 | (6.0) |
| P6 | I avoid crowded places (e.g., restaurants, religious gatherings, bars, etc.) as much as possible. | 804 | (47.1) | 480 | (28.1) | 367 | (21.5) | 57 | (3) |
| P7 | I avoid using public transportation. | 936 | (54.8) | 406 | (23.8) | 298 | (17.4) | 68 | (4.3) |
| P8 | I clean and disinfect surfaces that are frequently, touched (door handles, faucets and phone screens). | 406 | (23.8) | 343 | (20.1) | 502 | (29.4) | 457 | (26.8) |
| P9 | I avoid going out when I have cough or fever. | 822 | (48.1) | 436 | (25.5) | 381 | (22.3) | 69 | (4.0) |
| P10 | I visit the nearest flu clinic when I have fever or cough. | 949 | (55.6) | 292 | (17.1) | 327 | (19.1) | 140 | (8.2) |

## Discussion

In this study, we assessed KAP towards COVID-19 preventive measures among adult population of Bhutan. The knowledge on COVID-19 preventive measures was good and the majority held positive attitudes and good practice.

The knowledge about COVID-19 preventive measures was good in the majority. Similar findings with good knowledge about COVID-19 were reported among college students (74%) and medical students (98.4%) in Bhutan [15, 16]. Despite the difference in the study population, high knowledge score was reported in China [1, 17], Cameroon [18], Saudi Arabia [19, 20], and Malaysia [21]. However, studies conducted in Nepal [22], Bangladesh [23], Lebanon [24] and Malawi [25] reported of low knowledge score which was attributed by difference in background, sample characteristics and period of data collection.

The common sources of information were television, family and friends and social media. This is similar to surveys conducted in Bhutan, India and China where majority reported of getting COVID-19 related information from television and various social media platforms [16, 17, 26]. Television and social media have demonstrated effectiveness in reaching to the masses with information on COVID-19 preventive measures and government policies surrounding travel restrictions and mandatory quarantine [27].

The high rate of knowledge towards COVID-19 preventive measures among the participants is due to the wide initiatives; nation-wide lockdown, intensive disease surveillance, public exposure to the information taken by the government and media for educating public about COVID-19 from the start of the outbreak.

Young people (<30 years) were more likely to have good knowledge, which could be due to more exposure and easy accessibility of information on various social media platforms. However, low knowledge score was reported in government servants despite them being the more qualified group amongst all.

Concerning attitudes, the vast majority of the respondents held positive and optimistic attitude toward the COVID-19 preventive measures. Optimistic attitudes and high confidence towards the control measures of COVID-19 may have resulted from the government's unprecedented actions and quick response in adopting best global practices to protect the citizens and ensure their well-being and regular information about COVID-19 being updated in various mass media platforms. Mandatory quarantine for 21 days was initiated for travelers returning from third countries, 7 days for those travelling from high-risk areas to the low-risk areas, and lockdown of particular areas where community transmission was reported [15, 27]. In addition, mass gatherings were restricted and face mask are mandatory when in public places.

Although Bhutanese population showed high knowledge and optimistic attitude towards COVID-19 preventive measures on the contrary, their levels of practice was comparatively low. Our findings showed that knowledge of the participants on COVID-19 preventive measures were high on many items; for example, majority of the respondents (97.1%) knew that crowded places should be avoided; however, only 75.2% of the respondents practiced this particular preventive measure. In the current study, almost all the respondents stated that they used face mask while going out which might be attributed by mandatory use of face mask imposed by the government and strict monitoring of public compliance. However, only 43.9% of the respondents practiced surface disinfection of the frequently touched surfaces such as door handles, faucets and phone screens. The poor practice could be due to the lack of awareness about the cleaning and disinfecting high-touch surfaces which can also reduce the risk of infection [28].

Social scientists, especially those in public health and health communication, are working to identify the levels of knowledge, attitudes and practices on COVID-19 among the public to

design cost-effective public health campaigns and education programs [21]. The findings of the current survey indicates that the government and related agencies can make proactive use of the existing media platforms to inform the general population on public health interventions, policies, awareness-raising, and health education in an event of future health challenges and emergencies.

## Limitation

Although certain interesting finding were found in this study, several limitations should be acknowledged. First, casual inferences cannot be made due to the cross-sectional study design. Second, due to unforeseen lockdown in certain regions and travel restriction during the period of data collection, population from far reached areas could not be included in the study. Therefore, there is a limitation to the representativeness of the findings. Third, we acknowledge the possibility of reporting bias as the practice actions were self-reported. These can be addressed in future through observational research. Fourth, some may not have honestly reported due to social desirability.

## Conclusion

The majority had good knowledge on COVID-19 preventive measures. Television, friends and family members and social media were the common sources of COVID-19 information. The majority had positive attitude and practice towards adopting preventive practices.

## Supporting information

**S1 File. KAP COVID-19 preventive measures survey questionnaire Bhutan.**
(PDF)

**S1 Dataset. KAP COVID-19 preventive measures survey data.**
(XLSX)

## Acknowledgments

We thank Thinley Dorji (Central Regional Referral Hospital), Tshokey (Microbiologist, Jigme Dorji Wangchuck National Referral Hospital), Sangay Phuntsho (Vaccine Preventable Disease Program, Ministry of Health) and Tobgye (UNICEF, Bhutan), district administrators, enumerators and participants for their support.

## Author Contributions

**Conceptualization:** Tshering Yangzom, Tshering Cheki, Nirmala Koirala, Dipsika Rai.

**Data curation:** Tshering Yangzom.

**Formal analysis:** Tshering Yangzom, Dipsika Rai.

**Funding acquisition:** Tshering Yangzom, Tshering Cheki.

**Methodology:** Tshering Yangzom, Tshering Cheki, Nirmala Koirala, Dipsika Rai.

**Project administration:** Tshering Yangzom, Tshering Cheki.

**Resources:** Tshering Yangzom, Tshering Cheki.

**Supervision:** Tshering Yangzom, Tshering Cheki.

**Validation:** Tshering Yangzom.

**Visualization:** Tshering Yangzom, Tshering Cheki.

**Writing – original draft:** Tshering Yangzom, Tshering Cheki, Nirmala Koirala.

**Writing – review & editing:** Tshering Yangzom.

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
