## [Decision Letter · Decision Letter 0]

21 Jun 2022

PONE-D-22-15292Knowledge, attitudes and practices (KAP) towards COVID-19 preventive measures among Bhutanese population: A cross-sectional studyPLOS ONE

Dear Dr. Yangzom,

Thank you for submitting your manuscript to PLOS ONE. After careful consideration, we feel that it has merit but does not fully meet PLOS ONE’s publication criteria as it currently stands. Therefore, we invite you to submit a revised version of the manuscript that addresses the points raised during the review process.

We look forward to receiving your revised manuscript.

Kind regards,

Harunor Rashid, MD

Academic Editor

PLOS ONE

Journal Requirements:

2. Please provide additional details regarding participant consent. In the ethics statement in the Methods and online submission information, please ensure that you have specified what type you obtained (for instance, written or verbal, and if verbal, how it was documented and witnessed). If your study included minors, state whether you obtained consent from parents or guardians. Also, please clarify how you obtained consent from participants that were unable to read or write. If the need for consent was waived by the ethics committee, please include this information.

4. We note you have included a table to which you do not refer in the text of your manuscript. Please ensure that you refer to Table 6 in your text; if accepted, production will need this reference to link the reader to the Table.

Reviewers' comments:

Reviewer's Responses to Questions

**Comments to the Author**

1. Is the manuscript technically sound, and do the data support the conclusions?

Reviewer #1: Partly

Reviewer #2: No

Reviewer #3: Yes

Reviewer #4: Yes

2. Has the statistical analysis been performed appropriately and rigorously? 

Reviewer #1: No

Reviewer #2: Yes

Reviewer #3: Yes

Reviewer #4: Yes

3. Have the authors made all data underlying the findings in their manuscript fully available?

Reviewer #1: Yes

Reviewer #2: Yes

Reviewer #3: Yes

Reviewer #4: Yes

4. Is the manuscript presented in an intelligible fashion and written in standard English?

Reviewer #1: Yes

Reviewer #2: No

Reviewer #3: Yes

Reviewer #4: No

5. Review Comments to the Author

Reviewer #1: Knowledge, attitudes and practices (KAP) towards COVID-19 preventive measures among Bhutanese population: A cross-sectional study

Thank you for the opportunity to review this paper. This study examines Knowledge, attitudes and practices towards COVID-19 preventive measures among Bhutanese population

Some comments are listed below:

Line 100, sample size. It is unclear how the authors came up with the sample size of 1720. My calculation of the sample size with 5% Margin of error and 95% CI and an estimated response rate showed 754. I recommend that authors include the correct sample size and add a statement on why they have increased the sample size. Something like “ to allow for disqualification of incomplete responses; we increased the targeted sample size to 1720”.

Sampling methods and data collection heading

This section should be joined with sample size as it does not have any information about data collection, it just provides another version of sample size justification which is totally different from what has been stated in the previous section.

Line 108, authors included the selection of 86 per district and changed the sample size collection criteria. This is confusing! Please stick with on sampling methods and justify its use.

The choice of Pearson correlation is for two continuous variables and as per the study variable classification it is categorical.

35% of the respondents are illiterate, how does this affect the results of the study considering someone else is completing the survey on their behalf.

Table 2, remove no responses as it does not provide any additional information. Maybe list this table as a graph with bars the sources of information.

Table 3, an option of unsure should have been included in the survey questions. Also, table 3 should include a chi test results

Table 4, should include a chi test results

Table 5, should include a chi test results

Line 213-220; Correlation among KAP section needs to be updated with further details as it is unclear and does not cover relevant information from the data presented.

Thank you and good luck with your submission.

Reviewer #2: Thank you for giving me the opportunity to review this manuscript.

I can see that you have did a great work in this manuscript titled "Knowledge, attitudes and practices (KAP) towards COVID-19 preventive measures among Bhutanese population: A cross-sectional study". However, I do have more comments to give you, which I will discuss briefly here:

- In title, "attitudes" could be changed to "attitude"

Abstract:

- In lines 19-20, background statements need to be more introductory about either COVID-19 or the role of Bhutanese in this pandemic. It is not preferred to highlight this " To the best of our knowledge, this is the first study performed to assess the knowledge, attitude and practice of Bhutanese population towards COVID-19 preventive measures" in this place. However, you may state the aim in a more comprehensive way.

- In line 22, you may delete this "using a mobile application EpiCollect5 for data collection" from the abstract.

- In line 23, " .....surveyed from all 20 districts," which sampling technique was used here? stratified sample? systematic stratified sample?. please clarify this.

- In line 24, you may delete this: "A four section questionnaire was developed, and validated." Each participant was scored for each KAP section. "

- In line 25, you repeat " 1708 respondents." This needs to be deleted as it is unclear. Try to focus on your findings and try to explain them clearly.

- lines 35-39, need to be rewritten in a more intensive way and presented properly. You may present the mean COVID-19 knowledge score, attitude, and practice accordingly.

Overall, the abstract needs to be rewritten in a more intensive way.

- Introduction:

- In lines 47 & 51, "The infection was first detected in December 2019," duplications need to be deleted.

- In line 54,  "Covid-19" try to unify it all over the manuscript as COVID-19.

- In line 54, "number of those infected has.." infected what? cases? people? Please clarify it.

- In line 69, "however instituted in the early months of 2020..", delete however.. what months? Please summarize and cite it.

- In line 72, how about other preventive measures? vaccine? % of vaccinated people? try to add these.

- lines 77-80 could be shorted and merged.

Methods:

- In line 89, change "nationals" to "people".

- study setting needs to be stated clearly;  e.g., which 20 districts? any cities? Why did you choose these 20 locations? "

- In sample size, what is the total population of Butane? How did you assume without a total targeted population number? Please try to correct this.

- Is Epicollect5 the only platform available? Did all the butane have it ? Why do you just use this platform?

- In line 125, "Hence, the questionnaire was tailor-made to best suit the Bhutanese population." How did you confirm this? Did you pilot the Survey? If so, how? Which language did you use? If you pilot, then what is the alfa Cronbach score? 

- in line 130, " assess the participants' awareness." How to assess participants' awareness or knowledge? Please clarify this and correct it.

- In lines 136 and 137, " fourth section is constituted of 10 items (P1-P10), with four-point Likert scale responses (always=4, never=1)". This for what? What exactly is this evaluation?

Result;

-line 156, "response rate of 99.3%" How did you calculate this? If there is a withdrawal and a refusal rate? try to state that and the reasons behind it.

-In lines 156–160, do these findings correlate with the Butanes census? Alternatively, there is no random nor stratified selection.

-In line 220, abbreviations for the tables need to be stated and a footer needs to be added.

- Table 6 needs to be re presented with all the details Where is the P-value? 

-Regression could be a better explanation of these results if you did it.

-Discussion;

- Add some relevant comparative studies to arguments you results as the following Studies https://www.frontiersin.org/articles/10.3389/fpubh.2020.00217/full

https://journals.plos.org/plosone/article?id=10.1371/journal.pone.0244925

- Add comparison of your findings with other studies conducted among countries near Bhutan.

- Limitation section needs to be added.

Reviewer #3: I appreciate the effort of researchers to do such a comprehensive KAP study and examined statistical correlation between each variables. My comments are as follows:

1. Methods, result and discussion

a. Under demographic characteristics: Good to give information regarding the total population demographics on married, younger than 40 years of age, occupational distribution, and rural vs urban population so we can see the sample representation of this study.

b. Practice of COVID-19 preventive measures- it is challenging to assume practice based on how study participants answered few questions. Usually there is inflation of actual practice when compared to an observed practice. Ideally, observation of practice should have been part of the study to confirm practice rather than simple survey. To support that, is there any study in the same community that researchers can site to coraborate the practice of hand hygiene and use of facemask? It is helpful to see those data.

c. Another objective indirect surrogate marker for good attitude and practice would have been vaccine acceptance rate. Assumption is in society with high KAP of COVID-19, actual vaccine acceptance will be higher. Adding this on the discussion and examining its statistical correlation with attitude and practice might give another direction of examination of the data.

Reviewer #4: The manuscript discusses the KAP about COVID vaccination among Bhutanese people. Upon reviewing the manuscript I find following issues that require to be addressed which in my opinion would make the manuscript better in readability:

1. The details of clinical presentation of the first case are irrelevant in the instant manuscript and could be omitted (lines 73-75)

2. Did educational status determine the level of knowledge, etc. That might need address.

3. Did the various collected variables correlate with the KAP about COVID vaccines. Normally these sociodemographic features factors do affect.

Also the effect of age should be teased out. Younger population has greater access to modern modes of communication and it may be worthwhile to look at the association, if any.

4. Some discrepancy between the numbers in table 5 and those depicted here (line 208-210) . Needs to be ironed.

5. Grammatical error (P8 table 5)

5. Some discrepancy in numbers needs to be clarified. For example 48% always avoid going out if they feel unwell with fever or cough. However in P10, 55.6% ALWAYS visit a flu clinic when confronted with such symptoms. The numbers do not match and might need a relook by the authors.

6. The scoring system is not clear and we need to know the score classification that was attributed as low, high or average. For my understanding 89% is fairly high.

6. PLOS authors have the option to publish the peer review history of their article (what does this mean?). If published, this will include your full peer review and any attached files.

Reviewer #1: No

Reviewer #2: No

Reviewer #3: No

Reviewer #4: No

---

## [Author Response · Author response to Decision Letter 0]

5 Sep 2022

We thank you for your invaluable comments, suggestions and recommendations in our submission of the revised manuscript PONE-D-22-15292, titled “Knowledge, attitudes and practices (KAP) towards COVID-19 preventive measures among Bhutanese population: A cross-sectional study”.

---

## [Editor Report · Decision Letter 1]

7 Oct 2022

PONE-D-22-15292R1Knowledge, attitude and practices (KAP) towards COVID-19 preventive measures among Bhutanese population: A cross-sectional studyPLOS ONE

Dear Dr. Yangzom,

Thank you for submitting your manuscript to PLOS ONE. After careful consideration, we feel that it has merit but does not fully meet PLOS ONE’s publication criteria as it currently stands. Therefore, we invite you to submit a revised version of the manuscript that addresses the points raised during the review process.

We look forward to receiving your revised manuscript.

Kind regards,

Harunor Rashid

Academic Editor

PLOS ONE

Journal Requirements:

Additional Editor Comments (if provided):

Dear authors,

Thank you for submitting your revision.

The manuscript needs several more minor edits.

1. Please revise this statement in page 3, “According to the WHO, as of 11th April 2022, there were 497,057,239 confirmed cases of COVID-19 globally with a death count of 6,179.104 (5)”: please update the counts as of the date of submission, and revise ref 5, the death count was wrong.

2. Page 6, the last paragraphs: “..and the second, third and fourth section consisted of 28 questions: 12 knowledge-based, 6 attitude-based and 10 practice-based respectively’ are confusing; please state separately how many questions are there in the second, third and the fourth sections. We now understand the second section has 12 questions, so no need to repeat that in the subsequent paragraph. Avoid using ‘question’ and ‘item’ interchangeably, use one term consistently.

3. The scoring for the second section needs rethinking as it does not make logical sense. You gave 1 score for “don’t know”, 2 for “false” and 3 for “true”. False is the opposite of true, so if someone provides a false answer they deserve the lowest score, while ‘don’t know’ is an honest of deliberation of not knowing. In the society people who say they don’t know are safer than those who spread false information.

4. Some acronyms like ‘S-CVI’, ‘I-CVI’ and ‘CVR’ have not been spelt out on first use. Also in acknowledgement section hospital or department name like JDWNR hospital or VPDP need to be spelt out.
---

## [Author Response · Author response to Decision Letter 1]

12 Nov 2022

Dear Reviewers,

We would like to thank you for a thorough reading and constructive criticism of our manuscript titled “Knowledge, attitude and practice towards COVID-19 preventive measures among adults in Bhutan: A cross-sectional study”. Please find our response to reviewers’ comments in the table below. All the necessary changes and corrections in each section of the manuscript have been indicated in the “Revised Manuscript with Track Changes” file.

On behalf of my co-authors, I thank you for your consideration of this resubmission. We appreciate your time and look forward to your response.

Thanking you

---

## [Editor Report · Decision Letter 2]

18 Nov 2022

Knowledge, attitude and practices towards COVID-19 preventive measures among adults in Bhutan: a cross-sectional study

PONE-D-22-15292R2

Dear Tshering Yangzom,

We’re pleased to inform you that your manuscript has been judged scientifically suitable for publication and will be formally accepted for publication once it meets all outstanding technical requirements.

Kind regards,

Harunor Rashid, MD

Academic Editor

PLOS ONE

Additional Editor Comments (optional):

Please append the full questionnaire you used including demographic questions and sources of information. The current supplementary file you appended does not include the seven items on socio-demographic characteristics of participants plus the one on the sources of information.
---

## [Editor Report · Acceptance letter]

4 Dec 2022

PONE-D-22-15292R2 

Knowledge, attitude and practices towards COVID-19 preventive measures among adults in Bhutan: a cross-sectional study 

Dear Dr. Yangzom:

I'm pleased to inform you that your manuscript has been deemed suitable for publication in PLOS ONE. Congratulations! Your manuscript is now with our production department. 

Kind regards, 

on behalf of

Dr. Harunor Rashid 

Academic Editor

PLOS ONE